# Heme: The Lord of the Iron Ring

**DOI:** 10.3390/antiox12051074

**Published:** 2023-05-10

**Authors:** Vanessa Azevedo Voltarelli, Rodrigo W. Alves de Souza, Kenji Miyauchi, Carl J. Hauser, Leo Edmond Otterbein

**Affiliations:** Department of Surgery, Beth Israel Deaconess Medical Center, Harvard Medical School, Boston, MA 02215, USA

**Keywords:** heme, heme-oxygenase, traumatic brain injury, cancer, cardiovascular, sepsis

## Abstract

Heme is an iron-protoporphyrin complex with an essential physiologic function for all cells, especially for those in which heme is a key prosthetic group of proteins such as hemoglobin, myoglobin, and cytochromes of the mitochondria. However, it is also known that heme can participate in pro-oxidant and pro-inflammatory responses, leading to cytotoxicity in various tissues and organs such as the kidney, brain, heart, liver, and in immune cells. Indeed, heme, released as a result of tissue damage, can stimulate local and remote inflammatory reactions. These can initiate innate immune responses that, if left uncontrolled, can compound primary injuries and promote organ failure. In contrast, a cadre of heme receptors are arrayed on the plasma membrane that is designed either for heme import into the cell, or for the purpose of activating specific signaling pathways. Thus, free heme can serve either as a deleterious molecule, or one that can traffic and initiate highly specific cellular responses that are teleologically important for survival. Herein, we review heme metabolism and signaling pathways, including heme synthesis, degradation, and scavenging. We will focus on trauma and inflammatory diseases, including traumatic brain injury, trauma-related sepsis, cancer, and cardiovascular diseases where current work suggests that heme may be most important.

## 1. Molecular Structure, Physiological Function, and Toxic Effects of Free Heme

### 1.1. Heme Structure

Heme is a coordination complex consisting of an iron ion complexed with a porphyrin ring. When the iron atom is in the ferrous state (II), the complex is designated as either ferroprotoporphyrin or heme, and the molecule is electrically neutral [1]. When the iron atom exists in the ferric state (III), the complex is referred to as ferriprotoporphyrin or hemin, and the molecule carries a positive charge. Heme is hydrophobic, and there are several forms that exist in nature. The most common is heme b, which is found in the majority of hemoproteins including cytochrome b, hemoglobin, myoglobin, nitric oxide synthase, and cytochrome P450 [2]. It is also the precursor to heme a and c, which are the prosthetic groups in cytochrome a and c, respectively [1]. The structure of heme a is distinct from heme b in two main ways. First, heme has a hydroxyethylfarnesyl group at the C2 position instead of a vinyl side group. Secondly, it has a formyl group instead of a methyl side group at the C8 position. The transformation of the C2 vinyl group of heme b to the hydroxyethylfarnesyl group in heme a is carried out by the enzyme heme o synthase (HOS), which requires farnesylpyrophosphate as a co-substrate. The process of converting the C8 methyl group to the formyl group to produce heme a is catalyzed by heme a synthase (HAS), and necessitates molecular oxygen [2]. Heme c also differs structurally from heme b, which leads to distinct binding properties in forming hemoproteins because heme c covalently binds to proteins by two (rarely, one) covalent thioether bonds, whereas heme b binds to proteins non-covalently [3,4]. Due to its various structures, heme is able to interact with various proteins through covalent and non-covalent interactions, which are closely related to its functional diversity. As heme variants are metabolized by the heme oxygenases, the resulting bile pigment metabolites, biliverdin and bilirubin, which are generated with heme catalysis, may also exhibit unique chemical structures such as bilirubin a, which have specific functionality within the organism. In addition to these interactions, post-translational modifications of hemoproteins can also modify their functional capabilities [3].

### 1.2. Heme Physiology

The most commonly understood function of heme is its facilitation of the bulk transport of molecular oxygen (O_2_) from the atmosphere to the body cell mass by hemoglobin. Less commonly considered, but also of great and emerging importance, is heme’s role as a binder of gasotransmitters such as carbon monoxide (CO) or nitric oxide (NO), which ultimately influence the properties of the polypeptide within which the heme is contained as an active moiety. Indeed, molecular oxygen (O_2_) can also act in this capacity [2,4]. As such, heme is an essential molecule that is present in all cells, and functions as the prosthetic group for numerous hemoproteins such as hemoglobin, myoglobin, and mitochondrial cytochromes. These hemoproteins perform diverse biological functions, including oxygen binding and delivery to tissue, aerobic respiration, drug detoxification, and signal transduction [5]. Hemoglobin and myoglobin bind gasotransmitters with different affinities (CO > O_2_ > NO), whereas in cytochromes, heme is involved in electron transport and in facilitating the shuttling of electrons along the mitochondrial electron transport chain to generate ATP [4,6]. At least half of the known heme-containing proteins are in the cytochrome family, where they act as integral components of mitochondrial respiratory complexes II, III, and IV. These are crucial in cellular respiration and oxidative metabolism [7,8]. In catalases and peroxidases, heme functions to modulate the inactivation of hydrogen peroxide into water and oxygen [2,4]. Heme is also an essential moiety in cyclooxygenase and nitric oxide synthase [9]. Nitric oxide oxidizes heme iron to the ferric state, resulting in the formation of methemoglobin, which decreases the O_2_ carrying capacity of hemoglobin.

Heme is also important in the regulation of gene expression for numerous proteins such as globin, the heme biosynthetic enzyme 5’-Aminolevulinic acid synthase (ALAS), cytochromes, myeloperoxidase, heme oxygenase-1, Bach1, and the transferrin receptor [5]. Heme can regulate gene expression at the level of transcription, mRNA stability, splicing, protein synthesis, or post-translational modification [6]. The majority of the genes regulated contain heme response elements (HREs) located in enhancer regions that are modulated by intracellular heme levels [10].

### 1.3. Heme Synthesis

The regulation of heme synthesis and clearance is tightly regulated, and heme degradation is a critical physiologic response after tissue damage. Heme biosynthesis is initiated in the mitochondria by the first and rate-limiting enzyme, aminolevulinic acid synthase (ALAS). Heme plays an essential role in its own synthesis by regulating the expression of the two isoforms of ALAS: ALAS1 and ALAS2 [4]. In non-erythroid cells, heme directly regulates ALAS1 through a negative feedback mechanism in which heme suppresses the transcription, translation, and stability of ALAS1 mRNA [7,8,9]. This mechanism is critical for the maintenance of appropriate intracellular heme levels in non-erythroid cells, avoiding heme accumulation and the subsequent potential for oxidative damage. In bone marrow erythroid precursor cells, heme synthesis is exclusively dependent on ALAS2 activity, and its expression is controlled at multiple levels. At the post-transcriptional level, ALAS2 expression is regulated by iron availability, avoiding excessive heme production when iron stores are limited. The 5′ untranslated region of ALAS2 mRNA contains an iron responsive element (IRE) which interacts with iron regulatory proteins (IRP) 1 and 2. In iron-deficient cells, the binding of IRPs to IREs inhibits the translation of ALAS2 mRNA. In contrast, in iron-deficient states, IRP1 associates with an iron-sulfur cluster that prevents IRE binding. IRP2 interacts with the F-box and leucine-rich repeat protein-5 adaptor protein that complexes with SKP1-CUL1-F-box E3 ligase, thereby promoting IRP ubiquitination and the subsequent degradation by the proteasome [10]. Thus, during the differentiation of erythroid progenitors, increased cellular iron levels stimulate the translation of ALAS2 mRNA, leading to increased heme synthesis. This positive feedback mechanism allows the sustained production of heme necessary for hemoglobin synthesis during the differentiation of erythroid progenitors [4]. Interestingly, bilirubin has also been shown to modulate ALAS activity as an additional feedback regulatory mechanism. [11] In addition, ALAS contributes to increases in the expression of HO-1 in the presence of excess iron [12].

### 1.4. Heme Scavenging

Free heme released as a result of cellular or tissue damage rapidly complexes with plasma scavenger proteins that facilitate heme endocytosis with ensuant catalysis by the heme oxygenases. Hemopexin (Hx) is a serum glycoprotein (60 kDa) which binds to free heme with a high affinity. Heme is subsequently transported from the intravascular compartment to the liver and spleen for degradation by the reticuloendothelial system (RES) [13,14]. Hx is synthesized and expressed primarily in the liver, but it is also found in the brain and retina. Hx is an acute phase response gene that is activated by trauma, infection, stress, neoplasia, and inflammation. Many stimuli, but in particular hemolysis and inflammatory stimuli, upregulate Hx synthesis. Hx effectively reduces heme toxicity by 80–90% [15]. After binding free heme, the Heme–Hx complex binds to the LDL receptor related protein (LRP)/CD91 on plasma membranes. There, it forms endosomes which eventually mature into lysosomes where degradation occurs [16]. Some studies have suggested that Hx can be recycled as an intact molecule, but Hvidberg et al. suggest that most Hx is degraded in lysosomes [17].

### 1.5. Heme Degradation

Cytosolic degradation of heme proteins by lysosome results in the release of heme, which is transported to endosomes for protein catabolism and that contain heme oxygenases (Hmox1, HO-1 and/or Hmox 2, HO-2) for the metabolism of heme. Hemoglobin and myoglobin are the major sources of heme, with approximately 6−8 g of hemoglobin releasing about 300 mg of heme per day. Since free heme is not reutilized, it is degraded with the released iron stored as ferritin and eventually recycled into heme synthesis. HO-1 and HO-2 were first described in the late 1960s [18,19], and they have been identified in the endoplasmic reticulum [20], in plasma membrane caveolae [21], as well as in mitochondria [22] and the nucleus [23].

Each is the product of a different gene, and their regulation and expression differ greatly between cell and tissue types. HO-1 is highly inducible by a variety of danger signal stimuli including free radicals, hypoxia, ischemia-reperfusion, lipopolysaccharide, cytokines and nitric oxide as well as heme [24,25]. HO-1 is primarily expressed in the liver and spleen, where it functions to metabolize heme arising from senescent erythrocytes into bilirubin. However, under inflammatory and stress conditions, its expression can be induced in almost all organs [18,26]. In contrast, HO-2 as the constitutive isoform is expressed in the testes, neurons, glial cells, and cerebral vasculature [27]. HO-1 and HO-2 are predominantly localized to membranes of the smooth endoplasmic reticulum, where their action on heme results in generation of biliverdin, an atom of iron and a molecule of CO. HO-2 has also been shown to enter the nucleus and regulate gene expression [28]. The breakdown of heme into ferrous iron is carried out by the HO-CPR complex, which is located on the cytoplasmic face of the endoplasmic reticulum. Once the ferrous iron is released, it is exported into the bloodstream via FPN1. Ceruloplasmin oxidizes the recently released iron, which is then avidly bound by transferrin. This process provides the iron necessary for the generation of new erythrocytes. Biliverdin is further reduced to bilirubin by biliverdin reductase, and typically ends up accumulating as bile that aids digestion [29]. These bile pigments are powerful antioxidants, but they have also been shown to accumulate in the nucleus and regulate gene expression [30]. As iron is released, there is simultaneous upregulation of ferritin. Ferritin is an iron storage protein that protects against iron-mediated oxidative damage by scavenging the redox-active iron [31,32]. The inhibition of HO activity and thus iron release results in decreased ferritin levels [2]. Together, HO and ferritin rapidly shift iron from heme into the ferritin pool, where it is less available to catalyze the deleterious Fenton reaction and hydroxy radicals. Finally, the carbon monoxide (CO) released from heme breakdown is now widely accepted as a bioactive gasotransmitter akin to nitric oxide, with the ability to modulate hemoprotein function that would include guanylate cyclase, nitric oxide synthase and the mitochondrial oxidases. Importantly, in many instances, each product can mimic and substitute for the effects of HO-1 activity [15], but further detail on the three products is outside the scope of this review.

A small amount of heme is also degraded through HO-independent pathways. One such pathway involves GSH (reduced glutathione), which offers protection against heme-induced oxidative stress by degrading heme directly and scavenging heme-generated ROS [33]. Studies using intact erythrocytes indicate that GSH protects, in part by degrading membrane-associated heme [15]. It has been shown that interaction between the heme iron and the thiol group of GSH leads to a destruction of the tetra-pyrrole ring of heme and releases iron [34].

### 1.6. Heme Toxicity

Considering that heme exists normally as a component of intracellular proteins, the concentration of free heme in the plasma is very low, at an average concentration reported to be 21 ± 2 μM [11]. This low concentration is required because free heme is such a potent pro-oxidant and pro-inflammatory molecule that when it is present outside cells it is considered to be a Danger Associated Molecular Pattern (DAMP) molecule. As such, heme-related signaling can contribute to pro-oxidant cellular damage, the activation of innate immune responses, systemic inflammation, organ injury, and death. Also present in the circulation are the scavenger proteins haptoglobin and hemopexin (discussed above) that capture free hemoglobin and heme, respectively. Captured hemoglobin is transported to macrophages of the reticulo-endothelial system, where the complex binds to the scavenger receptor CD163 and is internalized. When the buffering capacity of plasma haptoglobin is exceeded, free hemoglobin is quickly oxidized in the plasma to methemoglobin with heme release for the subsequent elimination by hemopexin. This scavenging system is an essential defense mechanism, preventing heme from activating innate immune responses, and in scenarios where there is overwhelming cellular and tissue injury, hemopexin-mediated heme clearance may become rate limiting. This can leave free heme available to contribute to further inflammation and tissue injury, as described below [12].

### 1.7. Heme and Oxidative Stress

The pro-oxidant qualities of heme are driven in large part by the divalent Fe atom contained within the protoporphyrin IX ring that is redox-active and readily participates in Fenton chemistry [4,13]. These events generate the highly cytotoxic hydroxyl radicals that damage lipid membranes, proteins and DNA, as well as activating pro-inflammatory transcription factors [35,36]. Heme and iron can also lead to the disruption of membrane channels, particularly the IP_3_ and ryanodine receptor channels in the endoplasmic reticulum. This modulation of Ca^2+^ homeostasis can in turn modulate cell signaling cascades that ultimately result in cell dysfunction [37].

Because of its hydrophobic and lipophilic nature, heme readily accumulates in lipid membranes [13]. Once heme has intercalated into the membrane, the iron in it can catalyze the oxidation of cell membrane lipids, leading to the formation of lipid peroxides. The changes in membrane permeability that occur subsequent to these events ultimately lead to cell lysis and death [4,18]. Such ROS-mediated cytotoxic effects of heme can directly affect the function of various tissues and organs including the kidney, neurons, heart, liver and peripheral leukocytes [2]. In the kidney, heme promotes oxidative stress in renal epithelial cells by increasing lipid peroxidation and also stimulates local inflammatory reactions that contribute to renal failure [19]. In the CNS, hemoglobin released from red blood cells as a result of stroke, traumatic brain injury, or subarachnoid hemorrhage will act on cells that are exquisitely sensitive to the accumulation of heme. Thus, cell damage and the sudden release of heme into the extracellular milieu can even lead to remote, secondary injury, as has been observed in the brain [38]. Injury and stress are significantly enhanced if the heme oxygenases are absent in the brain [25,39]. Heme has also been found to be cytotoxic in the heart, causing the loss of cardiomyocyte integrity, sarcolemmal damage, and the release of cytosolic enzymes. In the liver, heme inhibits cytochrome P450-catalyzed reactions, which in turn can inhibit drug detoxification and enhance drug toxicity. In peripheral leukocytes, heme increases the expression of IL-2 and other cytokines that contribute to a cytotoxic response. Heme also binds TLR4, activating NF-kB signaling and increasing inflammatory mediator expression. The deleterious effects of heme have been implicated in the pathogenesis of numerous disorders, including sickle cell disease, malaria, atherosclerosis, vasculitis, and reperfusion injury, among others [2].

### 1.8. Hemolysis

As noted above, free heme promotes oxidative stress. This is particularly relevant to erythrocyte membrane stability, and free heme is known to be a potent hemolytic agent [20]. Thus, in circumstances where heme accumulates in red blood cell (RBC) membranes, it leads to the rapid loss of membrane integrity. There are several common clinical disease states such as thalassemia, sickle cell anemia, and malaria, where hemolysis regularly exposes tissues to large amounts of heme. These events commonly lead to vascular stasis, vaso-occlusive crises, endothelial cell activation, acute inflammation, and potentially pain [4,21]. Heme can also secondarily amplify hemolysis by membrane injury that stimulates potassium loss and the swelling of the RBC. Heme-induced hemolysis takes place in at least two phases. In the first phase, free heme impairs RBC membrane stability and the maintenance of ion gradients, leading to intracellular potassium loss. In the second phase, sodium and water enter the RBC, causing cellular swelling and eventual cell rupture [22].

### 1.9. Heme and Inflammation

Typically, once released from intracellular stores, heme is referred to as being ‘free’. That designation is highly controversial, however, since a hierarchy of heme scavengers is present. The highest affinity scavenger is hemopexin, followed by haptoglobin and then albumin, which have progressively lower affinities for heme [23]. Thus, one must be careful when referring to ‘free’ heme, as it likely only exists unbound to carrier proteins momentarily. For the purposes of this review, we equate free heme to that not bound to the high affinity carrier hemopexin and therefore more likely to be available to cause harm to cells and tissues. It remains unclear whether a heme-hemopexin complex versus a heme-albumin complex would carry different reactivity states with regard to pro-oxidant effects, but this is likely since there is a specific receptor for hemopexin and not albumin. Moreover, there are a significant number of studies that describe the hemopexin-independent effects of heme on cellular function. Therefore, once hemopexin is exhausted, the heme-albumin complex is likely responsible for initiating pro-inflammatory and DAMP-dependent effects.

It is now well-accepted that free heme can activate nuclear factor-κB (NF-κB) and MAPK pathways through toll-like receptor-4 (TLR4)/MD2 signaling in macrophages [24]. Myeloid differentiation protein 2 (MD2) is the accessory protein for TLR4, and is essential for the activation of TLR4 signaling in mediating inflammation [25,26]. A recent study showed that heme binds to the MD2/TLR4 complex and activates the MyD88 pathway, leading to the production of downstream effectors such as TNF [5]. TLR4 is a transmembrane protein expressed primarily in myeloid cells as part of a family of pattern recognition receptors, and is best known as the cognate receptor for LPS (lipopolysaccharide), a component of gram-negative bacteria. A series of elegant studies makes clear, however, that heme binding to TLR4 can initiate similar pro-inflammatory effects [35,40]. Additionally, heme can bind soluble MD2 and mediate endothelial cell activation [27]. Furthermore, the genetic or pharmacological blockade of MD2 effectively inhibits heme-induced, MyD88-dependent cytokine expression [41]. Collectively, these data suggest that MD2/TLR4 inhibition could be a potential therapeutic target in clinical diseases such as β-thalassemia, sickle cell disease, and malaria, where heme release is directly involved in the pathogenesis [5].

Heme has also been shown to induce IL-1β through the activation of the NLRP3 inflammasome in human endothelial cells. Erdei et al. demonstrated that heme activated NLRP3 in endothelial cells in vitro and in vivo, as evidenced by the increased expression of IL-1β. The molecular mechanisms by which heme activated NLRP3 involved ROS generation, as well as non-Hb-bound heme. Further investigations are needed to explore whether heme is involved in other mechanisms contributing to NLRP3 inflammasome activation, such as K+ efflux or lysosomal destabilization [28].

Heme can also function as a chemoattractant to promote the recruitment of leukocytes, platelets, and red blood cells to the vascular endothelium [5,29,30]. Moreover, heme also binds nitric oxide, thus impairing vascular relaxation [4]. In vitro, when free heme is added to endothelial cells, increases in the expression of intracellular adhesion molecules (ICAM-1) including vascular cell adhesion molecule 1 (VCAM-1) and endothelial leukocyte adhesion molecules (E-selectin) are observed. Such activated endothelial adhesion molecules subsequently recruit leukocytes, including neutrophils, that further contribute to local inflammation. The exposure of human neutrophils to hemin increases the expression of Interleukin-8, indicating a putative molecular mechanism by which hemin induces chemotaxis [30] and the recruitment of leukocytes, resulting in inflammation and eventual cytotoxicity, cell damage, and organ injury. Interestingly, all are thought to be due in part to the heme-mediated generation of intracellular ROS and the subsequent downstream activation of transcription factor signaling pathways, such as Nuclear Factor-Kappa Beta (NF-κB). Heme-mediated oxidative stress has been shown to activate NF-κB signaling by directly binding to and activating the inhibitor of NF-κB kinase (IKK). Activated IKK then phosphorylates the inhibitor of κB (IκB), leading to its degradation and subsequent nuclear translocation of NF-κB [42]. Once in the nucleus, NF-κB regulates the expression of target genes involved in inflammation and cell survival. Activator Protein-1 (AP-1) is another transcription factor signaling pathway that is activated. Heme has been shown to activate AP-1 signaling by increasing the phosphorylation and activation of c-Jun N-terminal kinase (JNK), a member of the mitogen-activated protein kinase (MAPK) family. Activated JNK then phosphorylates c-Jun, leading to the formation of an active AP-1 transcription factor complex that regulates the expression of target genes involved in cell proliferation, differentiation, and apoptosis [43,44,45].

Regardless of the origins of heme, i.e., parenchymal tissue damage, hemolysis, or hemoglobinemia, scavenging occurs rapidly, with receptor internalization leading to increased intracellular concentrations. As this occurs, specific heme sensors in the cell lead to the rapid induction of heme oxygenase as the enzyme responsible for its catabolism into three products that include iron, biliverdin, and carbon monoxide [30]. There are two isoforms of HO: HO-1, which is the inducible form, and HO-2, which is expressed constitutively, but primarily localized to the brain and testes. A low concentration of heme with concomitant elevations in HO-1 is part of the acute phase response, and as such acts as an anti-inflammatory and cytoprotective molecule. The mechanism has been studied for decades, and is thought to be due not only to the elimination of cytotoxic heme, but even more so to the generation of the bioactive metabolites biliverdin and CO. Both of these are potent signaling molecules that regulate cell survival [33]. Thus, while HO-1 is induced to reduce increased heme burden, its activity is rate-limited when metabolizing heme [23,34]. An increase in HO-1 expression has been shown both preclinically and clinically to impart salutary effects across a broad spectrum of inflammatory conditions including ischemia/reperfusion injury, atherosclerosis, asthma, traumatic brain injury, cardiotoxicity, Alzheimer’s disease, and acute renal failure, among others [35]. Taken together, heme is tightly regulated because of the broad spectrum of effects it can elicit in the cell and tissue. This may explain the battery of scavengers and highly specific metabolic enzymes that exist to ensure that heme is sensed, processed, and ultimately used by the cell to facilitate damage control and the restoration of tissue homeostasis.

In addition to its heme transporting properties, Hx has also been reported to exhibit antioxidant properties in that it inhibits hemin-induced lipid peroxidation [46,47]. The LDR/CD91 receptor mediates the regulation of several genes, including HO-1, transferrin, transferrin receptor, and ferritin, by binding heme-Hx complexes. Additionally, this receptor triggers the activation of genes involved in cellular defense against oxidative stress, such as metallothioneins, c-Jun, RelA/NF-kB, and metal-regulatory transcription factor 1 (MTF-1) when bound to heme-Hx complexes [48].

Human serum albumin has been reported to inhibit the toxic effects of hemin by 50–60% [2]. However, in comparison to Hx, the heme-albumin complex has been reported to function more as a heme repository as opposed to a transporter for specific targeting to the RES [47]. Another protein reported to possess heme scavenging properties is lipocalin alpha 1-microglobulin (alpha-1-m). This protein is synthesized in the liver, secreted into the circulation, and subsequently equilibrates across vessel walls from the intra- to the extravascular compartments in all tissues. It has been described as a tissue housekeeping protein responsible for removal of and protection against harmful oxidants including heme. Alpha-1-m has been shown to bind heme at a molar ratio of 2:1, such that two heme groups are bound to each alpha-1-m molecule [49,50].

### 1.10. Heme in Trauma and Inflammatory Diseases

As described above, heme is a molecule intimately involved in numerous cellular processes that range from oxygen transport, radical detoxification, and ATP generation. While clearly essential, it can also be dangerous when released into the bloodstream in large amounts, where it can trigger inflammation, oxidative stress, and cell damage. Heme is also a potent stimulus for immune system responses, and can lead to the formation of reactive oxygen species that can damage proteins, lipids, and nucleotides, leading to tissue damage and organ failure. In certain diseases, such as hemolytic anemia, malaria, or ischemia reperfusion injury, heme can accumulate in excess, overwhelm scavenging defenses, and cause significant harm to the tissue and organs. It is therefore important to regulate the levels of heme in the body so as to maintain its beneficial effects while still minimizing its potential dangers. We detail examples where heme contributes to tissue damage and the mechanisms that have been studied towards understanding how it signals in the body after tissue injury below. This porphyrin iron ring has enormous broad biological importance in instances of traumatic injury, infection, cancer, and cardiovascular disease. Each of these disease states has independently been shown to be influenced by heme release and metabolism as it relates to cell survival and tissue repair. In each instance, we make the argument that heme flux is a significant regulator of cellular homeostasis and tissue function (Figure 1).

### 1.11. Heme in Traumatic Brain Injury

Traumatic brain injury (TBI) is the leading cause of death and disability related to trauma worldwide. Recent reports state a global incidence rate of almost 1000 cases per 100,000. TBI caused by motor vehicle accidents, military missions, and athletic events results in a high incidence in acute and chronic disabilities, post-traumatic stress disorder (PTSD), and death [51,52]. External mechanical forces generated during impact and tissue compression from hematomas or edema can rupture blood vessels, causing secondary hemorrhage that leads to cortical tissue injury. This can then be followed by secondary injuries that can be the manifestations of either further ischemia or of the activation of innate immune responses. Thus, the evolving current understanding is that the secondary lesion is not simply ischemic, but includes ischemia plus an immunologic component that amplifies the initial insult. Local inflammation can cause the excitotoxicity of local neuronal cells, blood-brain barrier (BBB) damage, nerve death, hydrocephaly, impaired mitochondrial function, and increased glutamate imbalance, which can cause neuronal cell death and can continue for many years [53,54]. Compared to the initiating injury, secondary injuries can result in the delayed but progressive and profound damage of brain function [55]. In fact, prevention and care of post-TBI secondary insult is at the core of clinical intervention and recovery, and can greatly influence long-term prognosis [56,57], highlighting the importance of novel immune diagnostic and therapeutic strategies [54,58].

TBI can induce both focal brain damage due to the cellular release of danger molecules, as well as diffuse axonal injury resulting from sudden rotational injury caused by a sudden impact. This results in more diffuse dysfunction that can alter consciousness or be responsible for “high level” cognitive functional impairments, such as loss of attention, memory, and executive function, all of which are commonly seen in TBI patients [59,60,61]. Hemoglobin released from damaged erythrocytes as well as hemoproteins released from neurons, glia and endothelial cells after injury can rapidly expose the brain to high concentrations of heme [62] and associated iron. These can further enhance inflammatory responses and contribute to additional immune-mediated tissue injury [63]. The management of free heme might therefore be a reasonable novel target in efforts to restore neurologic function, control further anatomic damage, promote tissue repair, and thus potentially diminish the long-term sequelae of TBI.

HO-1 and HO-2 respond differently to TBI, with HO-2 expression being maintained throughout the post injury interval [64], while HO-1 shows a clear time- and location-dependent induction [63,65]. Early studies demonstrated associations between the post-injury induction of HO-1, oxidative stress, and cytotoxicity in TBI models: increased HO-1 expression was found to be selective for activated glia or infiltrating macrophages, but not detectable in injured neurons over the acute 24-h post-injury period [66,67]. Increased HO-1 expression was observed in the glia of the cortex, hippocampus, and thalamus after severe TBI in lateral midline fluid percussion injuries (FPI), and these were significantly attenuated by the antioxidant N-acetylcysteine [68]. This suggests that iron release and HO-1 activation may play important roles in TBI-induced secondary CNS injury [62,63,69]. More recently, HO-1 has been studied in additional animal models of severe TBI, including weight drop, controlled cortical impact [CCI], stab wounds, and cold lesion injuries. There, increased HO-1 expression was localized to microglia and macrophages at the site of injury, correlating with the extent of the hemorrhagic lesion, and occurred without significant changes in HO-2 expression [70,71]. Increased HO-1 expression has also been detected in human microglia and macrophages up to 6 months after TBI [72,73]. Finally, TBI patients with high injury severity have been found to have increased HO in their cerebrospinal fluid (CSF) and long-term HO elevations in the affected neurons and glia of the basal ganglia [74].

The induction of HO-1 expression can also have a protective role after TBI that is partially associated with increased mitochondrial biogenesis mediated through an AMPK-PGC-1α-ERR pathway. HO-1 expression may also improve mitochondrial quality control in astrocytes [75], which in turn positively modulates neural stem cell (NSC) function and neurogenesis in the peri-injury region [76]. Indeed, experiments using HO-1 metabolites or inducers, such as carbon monoxide (CO) or the HO inhibitor tin protoporphyrin IX (SnPP) have shown the beneficial neuroprotective effects of HO-1 and neurogenesis after TBI [72,76,77].

Deficient HO-1 expression/activity can lead to an imbalance in brain iron content, which is associated with various psychiatric disorders, such as major depression, bipolar disorder, and autism [78,79,80], as well as with Alzheimer’s, Parkinson’s, and Huntington’s diseases [81,82]. Patients with the rare neurogenetic disease “neurodegeneration with brain iron accumulation” (NBIA) exhibit elevated iron accumulation in the basal ganglia, which results in a range of psychiatric symptoms such as dystonia, spasticity, and parkinsonism [83,84]. A prevalence of 1% in systemic iron overload was observed in one study of psychiatric outpatients, which was found to be associated with a high rate of bipolar affective disorder diagnosis (80%) [74,79]. Elevated iron levels, likely due to heme degradation, have also been noted in myelinated tracts throughout the brain and in a subset of myelin-associated oligodendroglia [74] in such patients. Damaged endothelial cells, microglia, astrocytes, oligodendrocytes, and neurons can also facilitate iron accumulation post-TBI due to mitochondrial dysfunction, imbalanced redox state, excessive ROS production, or the disruption of iron-trapping protein expression [72,85,86]. Recent advances in MRI technology suggest that even mild TBI can generate CNS iron deposition [63,87].

Increased tissue iron content after injury can also induce iron-dependent programmed cell death, known as ferroptosis, which has been shown to play an important role in nerve damage after TBI [88,89,90]. During ferroptosis, heme-iron interacts with hydrogen peroxide producing hydroxyl radicals, which cause lipid peroxidation and glutathione depletion [91,92]. It is important to highlight that nerves may be particularly vulnerable to ferroptosis due to their cellular organization into bundles of axons and dendrites, as well as to contain unusually large amounts of cholesterol and polyunsaturated fatty acids in the cell membrane [52].

The activation of microglia by heme binding to toll-like receptor 4 (TLR4) activates two parallel signaling pathways: the myeloid differentiation factor 88 (MyD88)-dependent pathway, and the TIR domain-containing adaptor-inducing interferons (TRIFs) pathway, where both can ultimately induce the activation of the nuclear factor κB (NF-κB) [93,94]. Microglia activated by heme may polarize toward M1 phenotypes (classical macrophage activation) rather than M2 (alternative macrophage activation) phenotypes. Thus, M1 microglia are pro-inflammatory, and secrete numerous inflammatory chemokines such as TNF and IL-1β. M2 microglia exhibit anti-inflammatory activities and are more involved in tissue repair. After TBI, TLR4 expression also increases, again polarizing microglia toward the M1 neuroinflammatory phenotype. This can lead to neuronal loss and behavioral impairment [55]. Indeed, persistent neuroinflammation caused by microglial activation after TBI can influence the spread of abnormal proteins through the brain, becoming a causative factor in neurodegeneration. Neuroinflammation is often located at the sites of axonal pathology and can often spread far from focal injuries over time. This is known as Wallerian degeneration and is an active process of retrograde degeneration of the distal end of an axon that occurs as a result of a nerve lesion [59,95,96,97,98]. The known roles of heme in processes including the circadian rhythm, hunger, and satiety, as well as memory and pain sensation, all support its role as a critical regulator of neural signaling and homeostasis [64,99]. Nonetheless, its role in the brain and peripheral nervous system after injury remains relatively unstudied.

### 1.12. Heme in Trauma-Related Sepsis

Traumatic injuries are the third most common cause of death in the United States, and among individuals under 45, it is the leading cause of death. In contrast to the declining rates of cancer and heart disease deaths, which decreased by 20% between 1991 and 2009 and 31% between 2000 and 2010, trauma-related fatalities increased by 23% from 2000 to 2010 [100]. Following injury, hyperinflammation is accompanied almost instantaneously by immune suppression. This, in combination with the underlying injury pattern and severity, are critical factors contributing to the development of secondary complications such as multiple organ failure (MOF) and death [101,102]. Thus, while TBI and hemorrhage cause most early (<24 h) deaths, MOF associated with nosocomial infection causes most deaths occurring days or weeks later [103,104]. Moreover, trauma patients who survive their initial injuries have a significantly increased susceptibility to infection at sites remote from the primary injury. This puts them at risk for critical illness and mortality [105].

The primary mechanism by which an injured individual is believed to become vulnerable to infections involves the release of the danger-associated molecular pattern (DAMP) molecules following tissue damage. There are over 20 known and characterized DAMPs, but heme is perhaps the most well-studied [105,106]. Additionally, as evolutionary endosymbionts, mitochondria share molecular similarities with pathogen-associated molecular patterns (PAMPs) that can trigger potent innate immune responses similar to DAMPs [107]. DAMPs and PAMPs can modulate cell function through cognate receptors and signaling cascades. Inflammatory DAMPs can trigger a cascade of events, including a cytokine storm, oxidant radical generation, and the mobilization of transcription factors, which can lead to the expression of stress response genes [105]. However, this early response can also result in immunosuppressive events, potentially increasing the host’s susceptibility to infection.

There are a large number of proteins that contain heme as a functional prosthetic group in these compartments, and thus levels can increase rapidly with cellular rupture [108]. Trauma-related sepsis is characterized by intra- or extravascular hemolysis, rhabdomyolysis, and extensive cellular damage that promotes the release of large quantities of hemeproteins [109]. Studies have shown that heme activates the NLRP3 inflammasome [110], contributing to poor outcomes when present alongside lethal bacterial sepsis [111]. Furthermore, our research indicates that increased HO-1 activation by bacteria and CO generation are crucial regulators of pathogen-induced NLRP3 activation and bacterial killing [112]. Multiple lines of evidence suggest that heme plays a role in the pathogenesis of severe sepsis. First, the mortality of global HO-1 deficient mice exposed to microbial infection was aggravated and associated with the accumulation of free heme in plasma. Second, administering heme to wild-type mice exposed to low-grade microbial infection was sufficient to trigger a lethal form of severe sepsis. Third, heme accumulates in the plasma of wild-type mice subjected to high-grade microbial infection. Finally, sequestering heme through hemopexin suppressed the development of severe sepsis in wild-type mice exposed to a microbial infection [108,109,110].

Fortunately, the robust scavenging systems described above clear extracellular free heme. In 2016, Martins et al. found that reduced levels of circulating hemopexin were associated with the lethality observed in severe sepsis in mice and septic shock in humans [113]. Similarly, our study showed that mice with injuries that were infected with bacteria and treated with hemopexin could clear bacteria to levels equivalent to infected mice without trauma [64]. The beneficial effect of hemopexin likely depends on the expression of HO-1 to break down heme complexed to hemopexin. This could explain why HO-1-deficient mice are highly vulnerable to immune-mediated inflammatory diseases, even when in the presence of hemopexin [114], including polymicrobial infection [115] or during a beneficial stress, such as exercise training [116]. The induction of HO-1 expression and the generation of the heme metabolite CO in response to stress induced in response to microbial infection suppresses the development of severe sepsis and polymicrobial infection in mice [115].

As noted above, cell surface TLRs comprise a class of evolutionarily conserved pattern recognition receptors (PRR) that detect Pathogen Associated Molecular Pattern molecules (PAMPs) derived from invading pathogens that modulate innate immune responses to pathogens. TLR2 recognizes Gram-positive PAMPs such as lipoprotein and peptidoglycan [111,117], and several studies have shown the importance of TLR2 in *Staphylococcus aureus* infection. TLR2-deficient mice are highly susceptible to *S. aureus* infection [117,118,119,120], and perhaps this explains why *S. aureus* are more aggressive after trauma [121]. Interestingly, mice subjected to liver crush injury were significantly more susceptible to *S. aureus* infection in the lung, which was associated with the downregulation of TLR2 expression in the bone marrow and circulating neutrophils (PMN). These findings demonstrated that liver trauma with elevations in plasma heme induce the downregulation of both TLR2 and TLR4 in leukocytes. This was also observed in circulating PMN in trauma patients. This supports the concept that trauma-induced heme release suppresses TLR2/4 receptors as a manifestation of acute immunosuppression after traumatic injury and supports the global concept that injury-derived heme release leads to a higher rate of both Gram-negative and Gram-positive infections observed in trauma patients [64,111]. Another potential mechanism is immune nutrition, where the modulation of immune activity occurs via interventions with specific micronutrients, such as Fe, which is essential for bacterial growth and survival [113]. Because heme is a valuable Fe source, and some bacteria are unable to synthesize heme de novo, an altered supply of nutrients that emerges as a result of tissue damage modulates the inflammatory response and potentially provides a source of nutrients to support bacteria. Thus, trauma indirectly becomes a source of bacterial growth factors.

### 1.13. Heme-Induced Injury in the Cardiovascular System

Heme plays a role in the physiological functioning of various cell types in the cardiovascular system, including cardiomyocytes as well as smooth muscle and endothelial cells. However, the release of Danger Associated Molecular Pattern (DAMP) molecules in response to sterile injury and inflammation in several types of cardiovascular disease is an important aspect of organ damage, and ultimately contributes to dysfunction. Sickle cell and other hemolytic anemias frequently result in secondary pulmonary hypertension due to endothelial dysfunction [122]. One of the reasons is that these patients present increased circulating levels of ferrous heme and hemoglobin, which scavenge nitric oxide (NO), a potent vasodilator [123]. Moreover, increased plasma heme levels cause a reduction in the availability of nitric oxide (NO), which can lead to platelet activation and thrombosis [124]. This decrease in NO levels results in the activation of endothelial cells, characterized by an increase in the expression of adhesion molecules, including intracellular adhesion molecule 1, vascular cell adhesion molecule 1, and E-selectin. The changes in adhesion molecule expression on the surface of endothelial cells promote platelet adhesion and aggregation, suggesting that heme plays a role in platelet activation by scavenging NO [125]. Additionally, cell-free heme generates reactive oxygen species (ROS), further contributing to endothelial activation [125].

Excess heme can contribute to cardiovascular pathology by inducing smooth muscle proliferation. Hemolysis caused by abnormal blood flow can release heme [126], activating NADPH oxidase, a significant source of ROS in the cytoplasm. This activation has been demonstrated to cause smooth muscle proliferation, a pathological feature of atherosclerosis and hypertension [127,128]. Moreover, the inhibition of HO-1 can further potentiate smooth muscle cell proliferation [129]. In contrast, HO-1 upregulation by heme results in the production of CO, which imparts antiproliferative effects in part through blocking T-type Ca^2+^ channels [130]. The induction of HO-1, however, does not fully reverse smooth muscle cell proliferation in pathological conditions. Alternatively, reducing heme, such as with hemopexin, is another approach to limiting smooth muscle proliferation and vascular stenosis in diseases such as atherosclerosis or hemolytic anemia.

In addition to endothelial dysfunction, platelet aggregation, and abnormal smooth muscle proliferation due to increased levels of systemic heme, there is an association between hemin and myocardial damage [131,132,133]. Other studies have suggested that heme exposure impacts human cardiomyocyte morphology and contractile function by decreasing Ca^2+^ sensitivity (pCa50) [134]. In mice with hemolysis, free heme accumulates in the heart and induces ROS production, leading to alterations in Ca^2+^ homeostasis and a decrease in systolic function [135]. Experiments using heme-treated adult rat cardiomyocytes also demonstrate a significant reduction in systolic Ca^2+^ transient amplitudes [136]. One suggested mechanism is that heme-mediated ROS directly modify Ca^2+^-handling proteins, such as the ryanodine receptor-2 (RyR2) and sarcoendoplasmic reticulum Ca^2+^-ATPase 2a (SERCA2a) [137]. Furthermore, heme may activate intracellular stress kinases, such as calcium/calmodulin-dependent protein kinase II (CaMKII) [138], which in turn phosphorylates RyR2 and exacerbates Ca^2+^ mishandling. In the last few years, the role of iron in cardiac disease has garnered significant interest. Iron maintains cellular viability and function by contributing to oxidative phosphorylation, antioxidant enzyme activities, ribosome biogenesis, and oxygen storage and delivery [139]. Heme and iron–sulfur (Fe/S) cluster proteins account for most of the iron in the heart. Intravenous iron has been suggested to provide clinical benefit in iron-deficient patients with chronic systolic heart failure [140,141]. However, free iron is a highly reactive metal that catalyzes the production of toxic hydroxyl radicals through the Fenton reaction from less reactive species such as hydrogen peroxide and superoxide anion [142]. Several lines of evidence suggest that cardiac dysfunction is a hallmark of iron-overload diseases such as thalassemia, sickle cell anemia, primary hemochromatosis, or iron supplements, leading to cardiac iron overload [143,144,145]. In thalassemia and sickle cell anemia, red cells and their metabolites that appear in the circulation are degraded by the liver and spleen. This degradation releases excessive iron in the form of non-transferrin bound iron (NTBI). NTBI is an unstable combination of excessive iron ions [146]. Thus, the primary reason for reduced cardiac function in these patients is non-transferrin-bound iron (NTBI) rather than hemoglobin. Cardiomyocytes take up the Fe^2+^ from NTBI through bivalent transporters such as the L-type Ca channels, ZIP8, and ZIP14 in the heart, which may result in increased oxidative stress. This, in turn, can lead to a decrease in SERCA2 function and impaired contractility. Moreover, lipid peroxidation occurs within the mitochondrial membrane, leading to energy depletion, mitochondrial DNA damage, and dysfunction [147,148]. Theoretically, excessive intracellular Fe^2+^ catalyzes a cascade of lipid oxidation, which leads to ferroptosis [149]. Interestingly, Menon et al. revealed that increased heme present in sickle cell anemia upregulates HO-1, which in turn drives cardiomyopathy through ferroptosis. Moreover, the authors showed that the induction or inhibition in HO-1 activity decreased or increased cardiac ferroptosis, respectively [150].

Ferroptosis, an iron-dependent form of nonapoptotic cell death, was first identified in 2012 [151]. To date, it has been reported in several cardiac pathologies, including doxorubicin (DOX)-induced cardiomyopathy, acute ischemia/reperfusion injury (IRI) [152], post-myocardial infarction, heart failure [153], atherosclerosis [154], and septic heart injury [155]. In acute IRI, post myocardial infarction, and early-stage heart failure, the presence of residual myocardial iron in the post-infarcted area was observed with a concomitant decrease in GPX4 activity, which resulted in the ferroptosis of cardiac myocytes. In 2019, Fang and collaborators showed that DOX, an important class of drugs for treating cancer which unfortunately promotes cardiovascular injury, upregulates the Nrf2/HO-1 pathway, mediating heme degradation and free iron release, and resulting in cell death [152]. Thus, new studies are focusing on heme-binding therapies as cardioprotective strategies, such as the use of exogenous hemopexin. Liu et al., showed that exogenous hemopexin may mitigate DOX-induced ferroptosis to confer cardioprotection, suggesting that hemopexin may represent a compensatory response to DOX-induced heme release that occurs with treatment [156].

Collectively, as with trauma, it has become clear that an imbalance in heme metabolism is also important in the development of cardiovascular disease. The release of excessive free heme leads to phenotypic changes in endothelial cells, macrophages, vascular smooth muscle cells, and cardiomyocytes through oxidative stress and other signaling pathways. Furthermore, heme scavenging using hemopexin shows the potential for treating heme-mediated cardiovascular disease, but more investigations are warranted to study the effects of heme on sterile injury and inflammation in the cardiovascular system.

### 1.14. Heme in Cancer

Cancer is a disease characterized by genetic instability that leads to the uncontrolled proliferation of tumor cells [157,158], and which relies on the ability of these cells to escape immunosurveillance [159]. Approximately 29% of cancer cases are associated with environmental factors, which include smoking, a sedentary lifestyle, and diet, even though the mechanisms underlining these factors are not completely elucidated [160]. In this regard, inorganic and organic iron (in the form of the iron-containing porphyrin heme) have been shown to play important roles in the tumor microenvironment (TME) and metabolism, sustaining tumor cell proliferation and survival [161,162,163].

The role of heme in cancer and tumor growth has been attributed to its iron atom rather than to specific functions mediated by the entire heme molecule itself, and most studies in this field have focused on exogenous dietary heme as a risk factor for cancer. As an iron coordinated porphyrin predominantly contained in red and processed meat in the form of hemoglobin and myoglobin, heme has been proposed as a key molecule contributing to tumorigenesis upon red and processed meat intake [164,165,166]. The high dietary intake of red meat has been associated with an increased incidence of esophageal, gastric, breast, endometrial, pancreatic, and lung cancers [167,168,169,170,171], while most studies on the relationship between dietary heme and cancer have been associated with colorectal cancer (CRC) pathogenesis [172,173,174]. Increased amounts of red/processed meat intake can reduce heme absorption by the small intestine, leading to an accumulation of heme in the large intestine over the long-term [175,176]. In the presence of elevated free heme, both ferritin and HO-1 are rate limiting, and cells begin to accumulate free heme and labile iron that then exert a variety of cytotoxic effects on intestinal mucosa, including ROS-induced DNA damage [177,178,179]. Furthermore, it has been reported that heme alters normal intestinal bacterial flora, especially by decreasing the number of gram-positive bacteria [180], leading to a state of dysbiosis (microbial imbalance or maladaptation) that exacerbates colitis and adenoma formation in mice [181], and is correlated with the increased incidence of CRC [182].

Another possible effect of heme on tumor cells is related to heme biosynthesis, but this remains heavily debated. Heme biosynthesis, and specifically ALAS1, is frequently shown to be up-regulated in cancer [183], while the repression of heme biosynthesis by the aminolaevulinic acid dehydratase (ALAD) inhibitor, succinyl acetone, was shown to reduce tumor cell survival and proliferation [184,185]. Conversely, protoporphyrin ferrochelatase (FECH), the terminal heme synthesis enzyme, is often shown to be downregulated in tumors compared to normal cells [183,186]. The increased heme biosynthesis observed by tumor cells could act synergistically with improved mitochondrial oxidative phosphorylation (OXPHOS), since most cancer types produce adenosine triphosphate (ATP) via OXPHOS [187]. In fact, the induction of heme biosynthesis by ALA increases OXPHOS capacity in human lung carcinoma cells [188]. In summary, there is compelling evidence that both heme import and export processes are increased in tumor cells, which suggests a mechanism for heme homeostasis, and/or a response to two different heme pools (intra- or extracellular) that modulates cancer cell function. Additional studies are required to fully elucidate the biological significance of these alterations in heme trafficking in cancer.

Other than having a direct effect on tumor cells, heme might also play a role in cancer by modulating the TME. All cell types found in the TME contribute to tumor progression, establishing an optimal environment for tumor cell survival and migration, metastasis formation, chemo-resistance, and the ability to evade the immune system [189]. Heme is involved in nerve outgrowth in the TME through its ability to regulate nerve growth factor (NGF) signaling [190], and the metabolism of some neurosteroids and neurotransmitters since heme is required for the survival of different types of neuronal cells [191,192]. In addition, the innervation of solid tumors has been described in the literature, and the sympathetic nervous system has been proposed as a potential cancer hallmark [193], playing a key role in the development and progression of cancer [193,194]. Interestingly, mutations in genes encoding heme synthesis and heme export proteins have been reported in neuronal degeneration diseases [159,195]. Local and systemic sympathetic activation can also influence other cells in the TME, activating stress-related targets such as HO-1. In fact, it has been shown that the activation of β-adrenoceptors (β-AR) by isoproterenol (non-selective agonist) strongly increases HO-1 expression in macrophages through PI3K/Akt activation and Nrf-2 regulation, which can be prevented by β-blocker treatment [196,197]. Despite the evidence suggesting that there is an interaction between HO-1, sympathetic activity, and cancer, this association is still poorly understood and needs further study.

While HO-1 is an inducible stress response gene that is upregulated as a cytoprotective mechanism in disease states [198,199,200], its role in cancer is unclear, as there are studies that show that both the inhibition and induction of HO-1 can reduce the tumor burden [201,202,203]. One likely explanation is differences in kinetics, cell-types, and the choice of animal models. High HO-1 activity in tumor cells, likely through its ability to generate CO, can enhance cell death in many cancer models [201,204,205,206]. HO-1-induced ferroptosis is thought to be a potential mechanism of action [207,208]. In contrast, HO-1 induction can also promote tumor cell proliferation and metastasis [204,206,209]. This occurs in part by inducing immunomodulatory effects [210,211] and suppressing lymphocytic activity through the activation of regulatory T cells (Tregs) [212]. More recently, the expression of HO-1 by tumor-associated macrophages (TAMs) was shown to promote transendothelial migration and metastatic spread [213], while the absence of HO-1 in the myeloid compartment enhances T cell proliferation and cytotoxic effects against the tumor [202].

Heme has also been reported to modulate the profile and function of TAMs, directing them towards a pro-inflammatory M1-phenotype, and is thus considered to be the principle macrophage phenotype responsible for tumor cell death [214,215,216]. Indeed, the exposure of TAMs to hemolytic red blood cells (RBCs), commonly observed in cancer due to RBCs leakage from abnormal tumor-associated vessels [217], induces the accumulation of intracellular iron, and a switch to an M1 phenotype [216]. These findings suggest that heme acts to prevent tumor growth and progression.

Finally, heme metabolism alters endothelial cell function and strongly affects angiogenesis during development. The heme biosynthetic pathway plays a critical role in supporting endothelial and vascular function [218,219]. Tumor-associated endothelial cells are major components of the TME [220], with a high, aberrant, and dysfunctional neo-angiogenesis potential, which favor tumor growth and metastasis [221,222]. Therefore, one can surmise that heme is involved in tumor-induced angiogenesis, collaborating to positively influence cancer progression. More studies are needed to better understand the role of heme in TME vascularization. Collectively, heme is well-described in the cancer literature, but remains under heavy debate. On the one hand, heme released as a result of ongoing inflammation and immunosuppression caused by tumor growth promotes further cancer growth, while on the other hand, heme is cytotoxic, so as it accumulates after trauma it can drive processes such as ferroptosis [190,212]. Figure 2 illustrates some of the principal effects of heme in the TME that encompasses the immune response, angiogenesis, and innervation.

## 2. Conclusions

Despite its essential physiological role, we have provided an overview of how acute tissue injury leads to the release of heme from intracellular stores, and (whether scavenged or through binding to cognate surface receptors) leads to further signaling and the generation of bioactive degradation products as it is metabolized by the heme oxygenases. The mere fact that these systems exist suggest that heme serves as a hemodynamic molecule where, in some instances, it is important in cellular function, but as with the muses, also has a detrimental side, which, while dangerous, may in fact direct critical signaling pathways such as those mediated by the Toll-like receptors or other heme import and export channels. Moreover, the release and resulting signaling that heme elicits may serve the cell and tissue to promote damage control after traumatic injury and direct appropriate tissue survival and repair. The effects of heme in the setting of TBI, cardiovascular disease, and cancer, while seemingly disparate, share commonalities with regard to tissue stress, inflammation, and cell survival in the context of heme. It is more than possible that given the extensive handling processes in the body, including scavengers, receptors, and metabolic enzymes, heme may in fact be a soluble mediator much like a chemokine, a growth factor, or a neurotransmitter. More studies are needed in order to better elucidate the diverse effects of heme in physiologic and pathophysiologic scenarios, especially those related to homeostasis, inflammation, and innate immune responses.

With a better understanding of the mechanisms involved in heme regulation and metabolism, there is the potential to support novel therapeutic interventions, particularly given the breadth of scenarios where heme is elevated. Utilizing heme as a diagnostic to assess the severity of injury in the plasma of patients across different disease states may prove incredibly useful for optimal treatment strategies. The use of hemopexin and HO-1 as therapeutic agents to scavenge and metabolize free heme are certainly promising, with hemopexin entering clinical trials and HO-1 inducers such as Nrf2 agonists being in preclinical development. The intravenous administration of recombinant hemopexin is thus a feasible modality by which to reduce heme toxicity, as described in the conditions discussed herein, and can be administered prophylactically or therapeutically. Although a more complex approach, development of new chemical entities designed to elevate HO-1 expression may also emerge as potential clinical interventions in diseases where heme toxicity is problematic. As examples, dietary and phytochemical compounds, as well as the administration of CO to patients, are possible candidates that have been shown to up-regulate HO-1. It is important to highlight, however, that in complex conditions such as cancer, the application of such therapies becomes much more challenging because increased HO-1 is associated with the promotion of tumor growth, as described above. Therefore, in such instances it may require cell type-specific strategies, since heme/HO-1 appears to have a dual role in the tumor microenvironment. Taken together, this review provides a comprehensive description of the physiologic and pathophysiologic functions of a simple yet remarkable ferrous iron containing ring that reigns over numerous cellular processes.

## Figures and Tables

**Figure 1 antioxidants-12-01074-f001:**
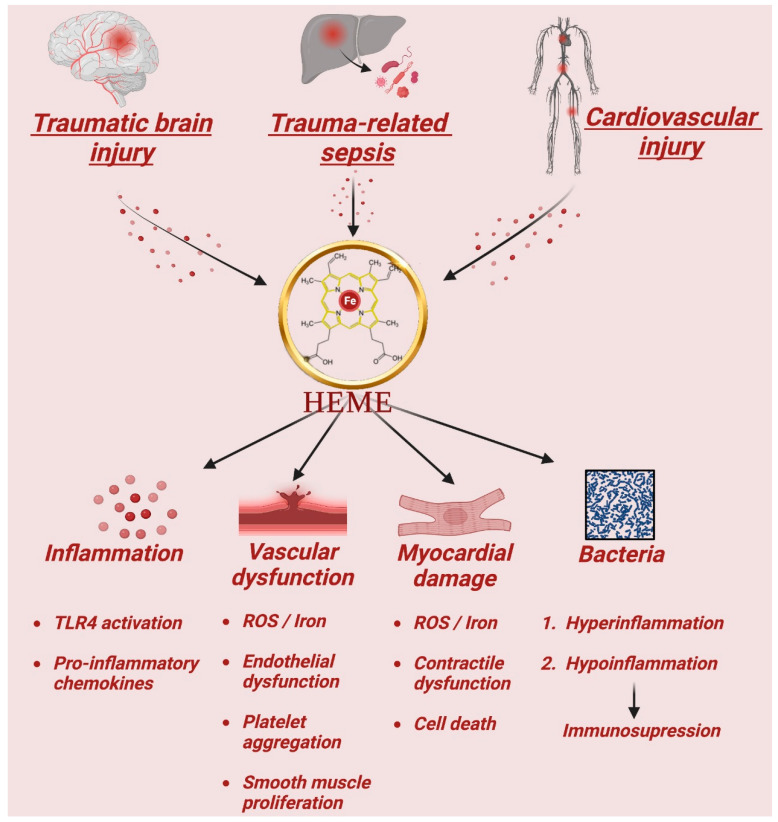
Heme “the lord of the iron ring” toxicity after large amounts are released into the bloodstream. Traumatic injury, infection, and disturbances in the cardiovascular system have all independently been shown to be influenced by heme release and metabolism as it relates to cell survival and tissue repair. Heme has toxic properties in each scenario due in large part to the catalytic active iron atom it coordinates, but heme also acts as a ligand for Toll-like receptors (TLR). After traumatic injury, high concentrations of heme contribute to an inflammatory response with additional heme-dependent tissue injury. In sepsis, the accumulation of heme as a result of tissue injury induces a prototypical cytokine storm, oxidant radical generation, transcription factor mobilization, and resulting stress response gene expression. However, tissue injury initiates a sequence of events that also leads to immunosuppression that increases the host’s risk of infection at barrier sites such as the lung. Disturbances in the cardiovascular system also promote the release of excessive heme, resulting in vascular dysfunction through the increased expression of adhesion molecules, endothelial activation, immune cell recruitment and platelet aggregation, as well as the consumption of nitric oxide that leads to vasoconstriction. Heme-injured myocytes exhibit morphological changes and oxidative enzyme loss, leading to a reduction in the contractile function of the heart. These effects highlight the importance of the tight regulation of heme levels in the body so as to prevent its harmful consequences on cell and tissue function.

**Figure 2 antioxidants-12-01074-f002:**
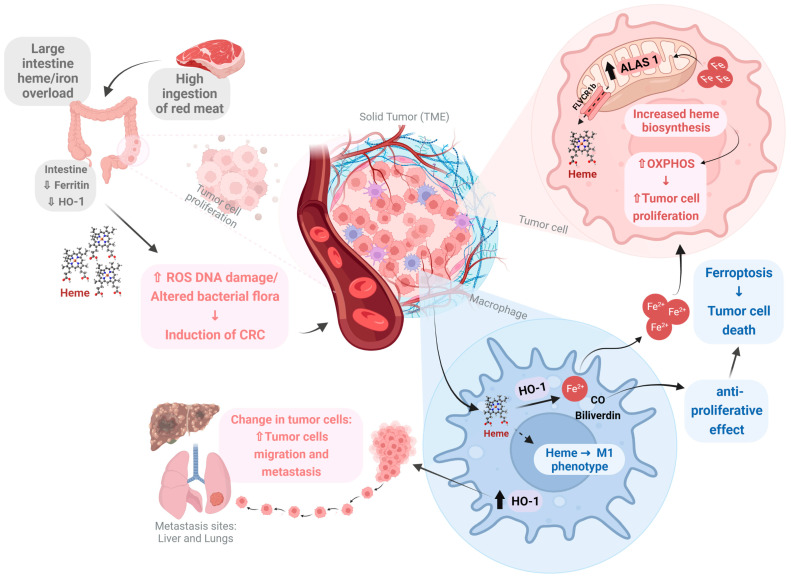
The dual role of heme in the tumor microenvironment. The modulation of ferritin and HO-1 expression in the setting of heme overload, such as the ingestion of red meat or acute tissue injury, induces DNA damage and increases the risk for colorectal cancer (CRC). In addition to the extracellular heme released by blood vessels irrigating the tumor microenvironment (TME), tumor cells can also increase heme biosynthesis while improving their mitochondrial bioenergetic response, including oxidative phosphorylation (OXPHOS), that augment survivability and proliferation. The increased HO-1 expression in tumor-infiltrating macrophages (TAMs) can promote tumor cell migration and metastasis. Conversely, TAMs can also switch to an antitumoral M1 phenotype in response to heme exposure in the TME. Moreover, heme metabolism by HO-1 in TAMs can induce tumor cell death in part by increasing Fe^2+^-mediated ferroptosis. Carbon monoxide (CO) can also impart potent anti-proliferative effects that may contribute to iron-mediated cell death. Thick arrows up indicate an increase, while thick arrows down indicate a decrease.

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
