# Peer review of "Heme: The Lord of the Iron Ring"

_antioxidants, 2023, doi:10.3390/antiox12051074_

Round 1

Reviewer 1 Report

In this manuscript, Voltarelli et al review important aspects of heme biology and further discuss the roles played by this molecule in some particular pathological conditions. The paper is well written and provides important information and hypotheses about the mechanisms through which heme in involved in oxidative stress and inflammation in such scenarios. However, I have some concerns that deserve special attention:

1 – Page 4, lines 141-143: This sentence is confusing. It sounds as if HO-1 and 2 were localized in endosomes, while it is mentioned later that they are attached to the membranes of endoplasmic reticulum (also, HO-1 can also localize to the nucleus where it can regulate gene transcription, which should be mentioned).

2 – Page 4, lines 177 – 181: Authors should provide a more detailed description of the process through which GSH promotes heme degradation. Moreover, the authors should reference the exact studies that characterized such mechanisms.

3 – Page 5, line 207: Authors describe NRF2 as a pro-inflammatory transcription factor, while in reality, NRF2 is as a master transcription factor that regulates antioxidant responses (which usually have anti-inflammatory properties).

4 – Page 6, 7, lines 290-292: Authors should provide further details of how heme modulates these NF-κB, AP-1 and SP-1 signaling pathways.

5 – Page 9, lines 435-437: This sentence is confusing and should be rewritten in a clearer way. Do the authors mean ferroptosis is more immunogenic than apoptosis? Please elaborate.

6 – In several parts throughout the text the authors have used other reviews to reference important specific information (for example, refs. 9, 30, 203, 204, 213, 215). Authors must whenever possible, refer to the original studies that described the specific phenomena. Please revise the manuscript to include the original citations where appropriate.

Reviewer 2 Report

The authors have provided a comprehensive and well-written description of the role and functions of heme in the manuscript. The depth of the analysis and the clarity of the writing make this review a valuable contribution to the field that resume the role of heme on trauma and inflammation processes. The references provided are also appropriate. 

Minor revision:

1. In my opinion, the authors could discuss, adding a table or a small paragraph, the potential therapeutic implications of heme modulation. For example, they could explore the potential of heme as a target for drug development in the diseases listed in the review, where heme metabolism is dysregulated.

Round 2

Reviewer 1 Report

The authors have responded to all of the concerns satisfactorily and improved the manuscript accordingly to the suggestions.